# Single-Molecule Real-Time Sequencing of Full-Length Transcriptome and Identification of Genes Related to Male Development in *Cannabis sativa*

**DOI:** 10.3390/plants11243559

**Published:** 2022-12-16

**Authors:** Hui Jiang, Ying Li, Mingbao Luan, Siqi Huang, Lining Zhao, Guang Yang, Gen Pan

**Affiliations:** 1Institute of Bast Fiber Crops, Chinese Academy of Agricultural Science, Changsha 410205, China; 2State Key Laboratory Breeding Base of Dao-di Herbs, China Academy of Chinese Medical Sciences, Beijing 100700, China

**Keywords:** *Cannabis sativa*, SMRT sequencing, male development

## Abstract

Female *Cannabis sativa* plants have important therapeutic properties. The sex ratio of the dioecious cannabis is approximately 1:1. Cultivating homozygous female plants by inducing female plants to produce male flowers is of great practical significance. However, the mechanism underlying cannabis male development remains unclear. In this study, single-molecule real-time (SMRT) sequencing was performed using a mixed sample of female and induced male flowers from the ZYZM1 cannabis variety. A total of 15,241 consensus reads were identified, and 13,657 transcripts were annotated across seven public databases. A total of 48 lncRNAs with an average length of 986.54 bp were identified. In total, 8202 transcripts were annotated as transcription factors, the most common of which were bHLH transcription factors. Moreover, tissue-specific expression pattern analysis showed that 13 MADS transcription factors were highly expressed in male flowers. Furthermore, 232 reads of novel genes were predicted and enriched in lipid metabolism, and qRT-PCR results showed that *CER1* may be involved in the development of cannabis male flowers. In addition, 1170 AS events were detected, and two AS events were further validated. Taken together, these results may improve our understanding of the complexity of full-length cannabis transcripts and provide a basis for understanding the molecular mechanism of cannabis male development.

## 1. Introduction

Cannabis is generally a dioecious species that includes both female and male plants [1]. However, in terms of appearance, the flowers of the female and male plants differ significantly. Female flowers develop from female inflorescences located at the axils of the leaves and lateral branches, as well as at the apex. The male flowers comprise an androecium of five short-stalked stamens enclosed by a perianth with five sepals. Moreover, the flowering times of female and male individuals in the same variety also differ, and male flowers tend to bloom earlier than female flowers [2,3].

Hemp is a multipurpose crop that is valuable for the production of fibers, cannabinoids, seeds, and oils [4]. Cannabinoids are unique phenolic substances found in cannabis, and there are approximately 100 types of cannabis [5]. Female flowers are the main sinks for accumulating large amounts of cannabinoids, such as cannabidiol (CBD), Δ-9- tetrahydrocannabinol (THC), and cannabigerol (CBG) [6,7]. Besides cannabinoids, terpenes, which also confer medicinal properties, are produced in floral tissues too [5]. Male plants are known to produce finer fibers than females [8]. Owing to the multifaceted nature of hemp products, a suitable proportion of males and females is particularly important during production. For example, if the purpose is to produce CBD and harvest seeds, the proportion of female plants in the population should be increased to gain high CBD and seed yield, whereas if the main purpose is to use fiber, the proportion of male plants should be increased to get more fiber with good quality. Therefore, considering its importance in determining the qualities of cannabinoids, terpenes, and fibers produced, understanding the molecular mechanisms of flower development in cannabis is essential for optimizing the production of its assets.

Many genes related to flower development have been extensively studied in some model species, such as *Arabidopsis thaliana*, snapdragon, and petunia, and the ABC(DE) model is an established model for mapping flower development in angiosperms [9]. In the ABC(DE) model, the biofunctions of different classes of genes are associated with the development of four floral parts: sepals, petals, stamens, and carpels [9]. For example, A genes are involved in the development of sepals and petals, B genes are responsible for the formation of petals and stamens, C genes have multiple functions that control the stamen carpels and ovules, D genes are only responsible for the development of ovules, and E genes control the development of sepals, petals, stamens, and carpels [10]. The key responsive genes reported previously in *Arabidopsis thaliana* include the A-class genes AP1 (APETALA1) and AP2 (APETALA1), the B-class gene PISTILLATA (PI), the C-class gene AGAMOUS (AG), the D-class genes FBP7 (Floral Bind 7), FBP1 (Floral Bind 1), and the E-class gene SEPALLATA (SEP) [11,12,13]. Among these five classes, the E-class protein plays an essential role in binding the other four class proteins to regulate the development of various floral parts. In rice, the ABC(DE)-class genes are mainly composed of MADS-box gene family members, including four genes under class A, three genes under B-class, two genes under C-class, two genes under class D, and six genes under E-class [14,15]. These findings indicate that MADS-box genes play important roles in the regulation of flower development. However, the role of MADS-box genes in cannabis flower development remains largely unknown.

Currently, the application of modern genomics technologies has laid an important foundation for accelerating the cultivar development of cannabis; therefore, accurate and complete referencing and annotation has been pivotal in aiding molecular research, which includes gene/quantitative trait locus mapping, genome-wide association studies, gene cloning, and genomic selection of key traits [16]. To date, 12 different cannabis cultivars have been assembled, from which four cultivar genomes have been of chromosome-level [17], whereas only annotation of the female cs10 has been made available. However, owing to the genetic diversity of cannabis germplasm and its characteristic dioecism, distinct alleles vary among cultivars and intra cultivars.

Next-generation sequencing (NGS) technologies, represented by the Illumina platform, has accelerated the construction of genome and transcriptome resources [18,19]. However, considering the limitation of short-length reads, it is difficult to accurately reconstruct expressed full-length transcripts, predict splice isoforms, and analyze transcriptome diversity using NGS reads [20]. Third-generation sequencing technologies, including Pacific Biosciences single-molecule real-time (SMRT) and Oxford Nanopore Technologies nanopore sequencing technologies, can compensate for these limitations and obtain high-quality long-read transcripts due to their ability to sequence reads up to 50 kb [21]. Here, we used SMRT to sequence the transcriptome of the female ZMZY1 variety cannabis plant bearing a chemical-induced male flower and obtain high-quality long-read sequences, alternative splicing, and isoforms. Additionally, genes related to male development were identified. These results will enhance gene annotation and our understanding of the cannabis transcriptome and the molecular mechanism of male development.

## 2. Materials and Methods

### 2.1. Plant Materials and Treatment Conditions

Cannabis seeds of the ZMZY1 variety were sown in a greenhouse at the Institute of Bast Fiber Crops, Chinese Academy of Agricultural Sciences. The growth conditions in the greenhouse were as follows: long-day treatment (18 h light/6 h dark cycle for 24 h, 26 ± 2 °C); and when the plants entered reproductive growth, short-day treatment were performed (8 h light/16 h dark cycle for 24 h, 26 ± 2 °C). Once the plant enters the reproductive growth period, silver thiosulfate (Ag_2_S_2_O_3_, 2 mmol/L) was sprayed on the top of the main stem and lateral branches of the plant according to previously reported methods [22]. On the 25th day following the first spraying, female flowers, male-induced flowers, roots, leaves, and stems were collected, frozen immediately in liquid nitrogen, and stored at −80 °C for SMRT sequencing.

### 2.2. Library Preparation and SMRT Sequencing

Total RNA was extracted using an EASYspin Plus Plant RNA Kit (Aidlab Biotechnologies Co., Ltd., Beijing, China) according to the manufacturer’s instructions. After assessing RNA degradation and contamination on agarose gels, its quality and quantity were determined using a NanoDrop 2000 instrument (Thermo Fisher Scientific, Waltham, MA, USA). The total RNA from each sample was mixed at an equimolar ratio. Then, 1 μg of total RNA was prepared for cDNA synthesis using the SMARTer PCR cDNA Synthesis Kit according to the manufacturer’s instructions (Clontech, CA, USA). PCR amplification was performed using PrimeSTAR GXL DNA Polymerase (Clontech). The SMRTBell library was constructed using the SMRTbell^TM^ Template Prep Kit 1.0-SPv3 (Pacific Biosciences, Menlo Park, CA, USA). The quantification and purity of the library was measured using a Qubit 2.0 Fluorometer (Life Technologies, CA, USA) and a Bioanalyzer 2100 system (Agilent Technologies, Santa Clara, CA, USA). Finally, SMRT sequencing was performed on the PacBio Sequel platform by Shanghai OE Biotech Co., Ltd. (Shanghai, China).

### 2.3. Quality Filtering and Error Correction

Raw reads were pre-processed and filtered using the PacBio SMRT Link v6.0.0 analysis package. Circular consensus sequences (CCS) were generated from the sub-read BAM files. CCSs were classified into full-length (FL) and full-length non-chimeric (FLNC) reads according to whether the 5′ and 3’ primers and poly (A) tails were detected. After clustering the FLNC reads, the arrow algorithm was used to obtain high-quality isoforms, and the corrected high-quality isoforms were compared to the reference genome using GMAP software. Non-redundant transcripts were obtained from the alignment results using Cupcake-ToFU at an identity value of 0.85. Furthermore, the SQANTI2 software sqanti_qc2.py and sqanti_filter2.py packages were used for the correction and filtering of non-redundant transcripts. BUSCO v3.0.1 software was used to evaluate the integrity of the transcript dataset.

### 2.4. Functional Annotation

The non-redundant transcripts were searched against six databases, including the non-redundant (NR), eukaryotic ortholog groups (KOG), gene ontology (GO), Swiss-Prot, protein family (Pfam), and Kyoto Encyclopedia of Genes and Genomes Ortholog (KEGG) databases, using Diamond software (E-value < 1 × 10^−5^). Pfam was compared with the protein family model using HMMER software, and the family with the highest score was selected.

### 2.5. Identification of Coding Sequences, Non-Coding RNAs, and Transcription Factors

Non-redundant transcripts (from the NR, Swiss-Prot, and KOG databases, in descending order of priority) were compared with the above-mentioned library using BLAST (E-value < 1 × 10^−5^), and the coding sequences (CDS) of unmatched transcripts were predicted using ESTScan software. To identify long non-coding RNAs (lncRNA), transcripts with lengths greater than 200 bp and open reading frames (ORFs) greater than 300 bp were filtered out. The sequences annotated in the coding library were also screened out. LncRNAs were further screened to filter out transcripts with coding potential using four analysis tools, including the coding potential calculator (CPC), coding-non-coding index (CNCI), Pfam, and predictor of long non-coding RNAs and messenger RNAs based on an improved k-mer scheme (PLEK). NR transcripts were compared to the Plant Transcription Factor Database (PlantTFDB) using BLASTX (E-value < 1 × 10^−5^) to predict transcription factors (TFs).

### 2.6. Alternative Splicing Analysis and Validation

Based on the results of SQANTI analysis, Astalavista software was used to ascertain alternative splicing events. The main types of alternative splicing include intron retention (IR), exon skipping (ES), alternative 5′ splice sites (A5), alternative 3′ splice sites (A3), and mutually exclusive exons (MXE). To validate the detected AS events, two unigenes were randomly selected for validation using male-induced cannabis flowers (IMF). Primers were designed for selected unigenes using Primer 5.0 software (Appendix A). Total RNA from the male-induced flowers was extracted as described above. The Goldenstar^TM^ RT6 cDNA Synthesis Kit Ver.2 (TsingKe, China) and 2×Es Taq MasterMix (Dye) (Cowin, China) were used for reverse transcription and PCR assays, respectively. The PCR amplification conditions were as follows: 94 °C for 1 min, 94 °C for 30 s, 55 °C for 30 s, 72 °C for 1 min (30 cycles), and 72 °C for 3 min. PCR products were detected using 1% agarose gel electrophoresis. The FastPure^®^ Gel DNA Extraction Mini Kit (Vazyme, China) was used for gel recovery, and the *pEASY*^®^-Blunt Cloning Kit was used to ligate the recovered product to the B vector. This was sequenced to confirm whether the amplified sequences were consistent.

### 2.7. Real-Time Quantitative Reverse Transcription PCR Validation of Novel Genes

Three different tissues from ZMZY1 variety cannabis female flowers (FF), male flowers (MF), and male-inducing flowers (IMF) were used to validate the gene expression levels of novel genes enriched in lipid metabolism pathways. Total RNA was extracted as described previously. Reverse transcription and real-time quantitative reverse transcription PCR (qRT-PCR) were performed according to the instructions of the *Evo M-MLV* RT Premix for qPCR Kit (Accurate, Changsha, China) and SYBR^®^ Green Premix *Pro Taq* HS qPCR Kit (Accurate, Changsha, China). Primer 5.0 was used for primer design, the *eIF4a* gene was selected as the internal control, and 1% agarose gel electrophoresis was used to detect primers (Appendix A). Data analysis was performed using the 2^−ΔΔCt^ method.

## 3. Results

### 3.1. Single-Molecule Real-Time Sequencing

The qualified RNAs from different tissues of the ZMZY1 variety were equally mixed, and the SMRT sequencing was performed using the PacBio Iso-Seq platform to obtain the full-length transcriptome of cannabis. A total of 10.32 million subreads with an average length of 1679.95 bp were obtained from the polymerase reads after the adapter sequences were removed (Appendix A). After comparison and correction, a total of 347,387 CCS reads were generated, of which there were 290,886 FLNC reads, accounting for 83.74% (Table 1). The FLNC reads were clustered and then corrected using the arrow algorithm, and 32,635 high-quality isoforms were obtained, with an average length of 1941.68 bp (Appendix A). Comparing high-quality isoforms with the reference genome, 97.4% were mapped to the reference genome. Then, de-redundancy was performed using Cupcake-ToFU software, and 15,241 reads of non-redundant transcripts were obtained. After further correction and filtering using SQANTI software, 13,657 non-redundant transcripts were obtained for subsequent analysis. The mean length of the non-redundant transcripts was 1994.66, the maximum length was 7150 bp, and the minimum length was 212 bp. Finally, the integrity of the transcripts was estimated, and there were 51.1% reads (735) matched to the complete BUSCO database (Appendix A).

### 3.2. Coding Sequence and lncRNA Identification

A total of 13,540 coding sequences (CDS) were obtained using BLAST software, of which CDS length > 2400 bp accounted for only 9.35%, whereas the proportion of CDS with a length of 800–1600 bp was larger (6348, 46.88%) (Figure 1). LncRNAs are non-coding RNA molecules with a length of more than 200 bp. In this study, the numbers of lncRNAs identified in CPC, CNCI, Pfam, and PLEK database were 53, 50, 53, and 51, respectively (Figure 2). Moreover, 48 lncRNAs were simultaneously screened from four databases, with an average length of 986.54 bp.

### 3.3. Functional Annotation of the Cannabis Full-Length Transcriptome

The public database annotated 13,657 non-redundant transcripts, including NR (13,538), Swiss-Prot (11,587), KEGG (6025), KOG (8552), eggNOG (13,379), GO (10,896), and Pfam (12,844) (Table 2). Additionally, 4808 intersection genes had significant hits in all seven public databases (Appendix A).

A total of 13,538 non-redundant transcripts were annotated by comparing them with the NR database. The results showed that *Trema orientale* (52.84%), *Parasponia andersonii* (22.9%), *Morus notabilis* (12.59%), and *Ziziphus jujuba* (2.09%) were the largest distributed species, of which 0.53% non-redundant transcripts can be compared with non-redundant cannabis sequences in the NR database (Figure 3). Moreover, a total of 8552 non-redundant transcripts were annotated in the KOG database and divided into 25 functional classes (Figure 4a). Among them, the R (general function prediction only; 1508), O (posttranslational modification, protein turnover, chaperones; 1043), T (signal transduction mechanisms; 892), and G (carbohydrate transport and metabolism; 615) groups had the largest number. Based on the results of Swiss-Prot database analysis, the GO database enriched 10,896 non-redundant transcripts, which were assigned to three categories, namely: biological process, cellular component, and molecular function (Figure 4b). Cellular and metabolic processes were enriched in biological processes. Within the classification of cellular component, “cell” and “cell part” were enriched. In terms of molecular function, “binding” and “catalytic activity” were significantly enriched.

### 3.4. KEGG Enrichment Analysis

In total, 6025 non-redundant transcripts were annotated into 23 processes in the KEGG database, of which the highest percentage was classified as “carbohydrate metabolism” under metabolism, accounting for 18.24% of the total. Transcripts classified as “translation” under genetic information processing accounted for 15.87%, “signal transduction” under environmental information processing accounted for 16.12%, and “transport and catabolism” under cellular processes accounted for 9.21% (Appendix A). Carbohydrate metabolism and signal transduction were the highest among all categories (Appendix A). There were 201 pathways among all the 23 processes. Among the top 10 significantly enriched of the 201 pathways, 321 transcripts were enriched in the carbon metabolism pathway (Table 3).

As mentioned above, carbohydrate metabolism accounts for the highest proportion of all classifications (Appendix A). It is evident from Table 3 that the pathway of starch and sucrose metabolism was significantly enriched in carbohydrate metabolism. Moreover, the expression levels of genes involved in starch and sucrose metabolism pathways were analyzed in different tissues based on the transcriptome data of the cannabis monoecious strain USO14 (data unpublished). A total of 70 genes were involved in the starch and sucrose metabolism pathways, of which 20 genes were highly expressed in male flowers and 13 genes were expressed at low levels in male flowers (Figure 5). The highly expressed genes included *BAM1*, *GAUT*, and *BGL40* (Appendix A).

### 3.5. Identification and Analysis of TFs

As known, transcription factors play a key role in gene expression regulation. In this study, a total of 8202 transcripts were annotated as TFs and grouped into 58 categories. The TF family of C2H2 (407), ERF (482), MYB-related (624), NAC (584), and bHLH (820) were significantly enriched (Figure 6). The family number of bHLH was the highest among the 58 categories (Figure 6). Besides, in view of the importance of MADS genes in plant floral organ development [14,15], their transcription levels were further analyzed based on the recently obtained transcriptome data of ‘USO14′ monoecious cultivars in our lab (unpublished). There were 47 genes encoding MADS transcription factors, of which 13 were highly expressed in male flowers (Figure 7). These results indicate that the 13 MADS transcription factors with high expression in male flowers may be involved in the regulation of male flower development.

### 3.6. Novel Gene Prediction and KEGG Enrichment Analysis

Based on the existing genome structure annotation, 232 reads of novel genes were predicted in non-redundant transcripts, with an average length of 1797.98 bp (Table 4; Appendix A). To further investigate the molecular mechanism of male development in cannabis, novel genes were matched in the KEGG database, of which 48 transcripts were annotated to 46 pathways (Appendix A). As shown in Figure 8, metabolism was mainly present in four classes, of which eight and five pathways were enriched in lipid metabolism and carbohydrate metabolism, respectively. In addition, seven pathways were enriched in signal transduction. The enrichment results of novel genes were similar to the above KEGG enrichment results, with significantly enrichment in carbohydrate metabolism and signal transduction. However, lipid metabolism was significantly enriched in the novel gene annotation. Therefore, in order to further analyze the expression of novel genes in the lipid metabolism pathway in cannabis, three genes were randomly selected for qRT-PCR. The results showed that, compared with MF and IMF, the expression levels of *PLD1_2* and *KCR1* were higher in FF, whereas the expression level of *CER1* was lower. In particular, the expression trends of MF and IMF were consistent but showed differential expression with FF (Figure 9).

### 3.7. Identification and Validation of AS

AS allows a gene to produce multiple mRNA transcripts; therefore, a gene may produce multiple proteins through alternative splicing, which greatly increases protein diversity. In this study, a total of 1170 AS events were detected using the Astalavista software, including 138 alternative 5′ splice sites (A5SS), 325 alternative 3′ splice sites (A3SS), 151 exon-skipping (ES) events, 554 IR, and two mutually exclusive exons (MXE) (Figure 10a). The IR events were the most frequent of all AS events, accounting for 47.35%. To further validate AS events from the transcriptomes, we randomly selected two genes (PB.870 and PB.43929) for the PCR assay. As shown in Figure 10b, the band sizes of the fragments of the two genes were the same as those of the splice isoforms in the full-length transcriptomic data. All genes had two bands amplified in IMF that were identical to the putative splice isoforms. The gene fragments were cloned, and the sequencing data were aligned with the corresponding genes to further verify the splice junctions in AS events.

## 4. Discussion

RNA sequencing (RNA-seq) is an important technical tool for studying differentially expressed genes and gene structures. However, the limitation of its short sequencing read length makes it difficult to accurately obtain full-length transcripts from the assembly of short-length reads. Using third-generation sequencing technology, including SMRT of PacBio, allows direct de novo sequencing of the complete mRNA, which compensates for the read length limit [23,24,25]. Adal et al. obtained a total of 73,833 transcripts in a transcriptome analysis of the masculinization of female cannabis. The length distribution of transcripts ranged from 200 bp to 14,620 bp, whereas only 17% of transcripts were ≥1500 bp in length [11]. In this study, a total of 347,387 CCSs and 290,886 FLNCs were obtained and 13,657 non-redundant transcripts were obtained with an average length of 1994.66 bp (Figure 1, Table 1), which was longer than that in the previous study [11].

Carbohydrate metabolism plays an important role in pollen viability, especially starch and sucrose, which provide metabolic substrates for pollen during pollen development [26,27,28,29]. In this study, the classification of carbohydrate metabolism was significantly enriched in the KEGG database, and the starch and sucrose metabolism pathways (173) accounted for the largest proportion of carbohydrate metabolism (Table 3). Moreover, 20 genes involved in the starch and sucrose metabolism pathway were highly expressed in the male flower, including galacturonosyltransferase (*GAUT*), beta-amylase 1 (*BAM1*), and beta-glucosidase 40 (*BGL40*) (Figure 5, Appendix A). In the previous studies, *GAUT*, *BAM1*, and *BGL40* were reported to play an important role in pollen tube growth [30,31,32], anther development [33,34,35,36,37]. Thus, we speculated that *GAUT*, *BAM1*, and *BGL40* may be involved in the male flower development of cannabis, and their biological will be validated in the further study.

In comparison with RNA sequencing, third-generation sequencing technology can provide more information about specie transcriptom. This study identified 232 transcripts as novel genes, of which only 48 genes were annotated in the KEGG database. Among the 46 pathways, the number of pathways involved in lipid metabolism was highest. Three genes (*PLD1_2*, *CER1*, and *KCR1*) involved in lipid metabolism were found to be significantly differentially expressed between FF and male flowers (MF and IMF) (Figure 9). In some plant species, the *CER1* gene is involved in the pathway of cutin, suberine, and wax biosynthesis, and plays an important role in the pollen fertility of plants [38,39]. For instance, the downregulation of the *OsCER1* gene can lead to pollen sterility in rice [40]. Thus, *CER1* may act as an important role in the development of male flowers in cannabis.

In this study, a total of 48 lncRNAs were screened in the CPC, CNCI, Pfam, and PLEK databases (Figure 2). LncRNAs are considered key factors in plant transcriptional regulation and play important regulatory roles in plant growth and development, flowering, sex differentiation, male sterility, and biotic and abiotic stresses [41,42,43,44]. In rice, long-day-specific male-fertility-associated RNA (LDMAR) plays a critical role in normal pollen development under long-day light, and reduced LDMAR transcription leads to premature programmed cell death in developing anthers, which in turn leads to the production of photoperiod-sensitive male sterile lines [45]. Whether the lncRNAs identified in this study are involved in male development of cannabis requires further analysis and validation.

## 5. Conclusions

In this study, we used SMART technology to obtain full-length transcripts of female cannabis plants with induced male flowers. A total of 15,241 consensus reads, 48 lncRNAs, 1170 AS events, and 8202 TFs were identified, and the expression levels of 47 MADS TFs and 3 novel genes related to lipid metabolism were analyzed. This study may provide more useful information for optimizing genome annotation and structure and lay a foundation for understanding the molecular mechanism of cannabis male development.

## Figures and Tables

**Figure 1 plants-11-03559-f001:**
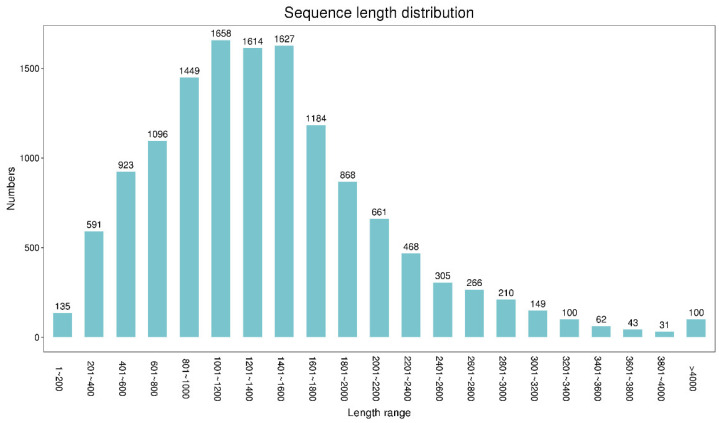
Sequence length distribution of CDS.

**Figure 2 plants-11-03559-f002:**
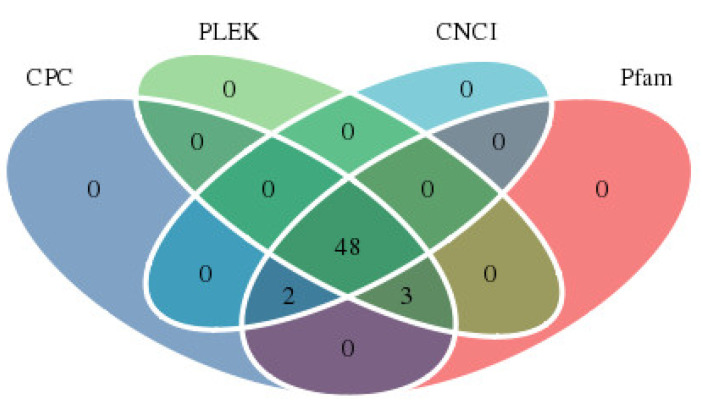
Statistics of lncRNA annotation in CPC, PLEK, CNCI and Pfam databases.

**Figure 3 plants-11-03559-f003:**
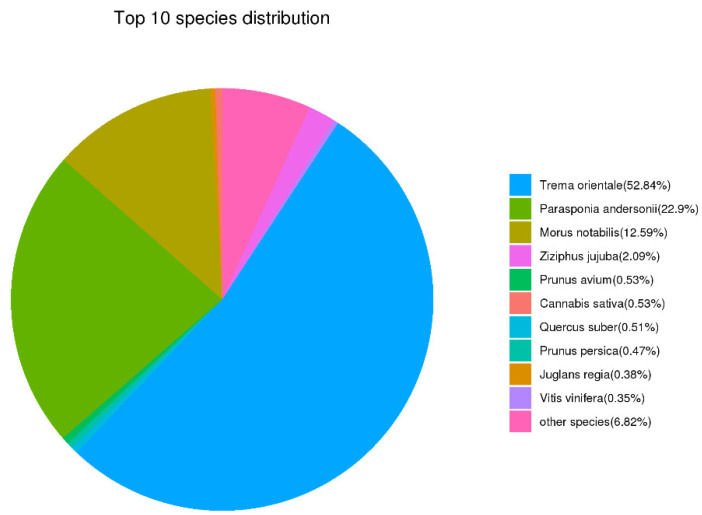
The classification statistics of non-redundant transcripts in NR database.

**Figure 4 plants-11-03559-f004:**
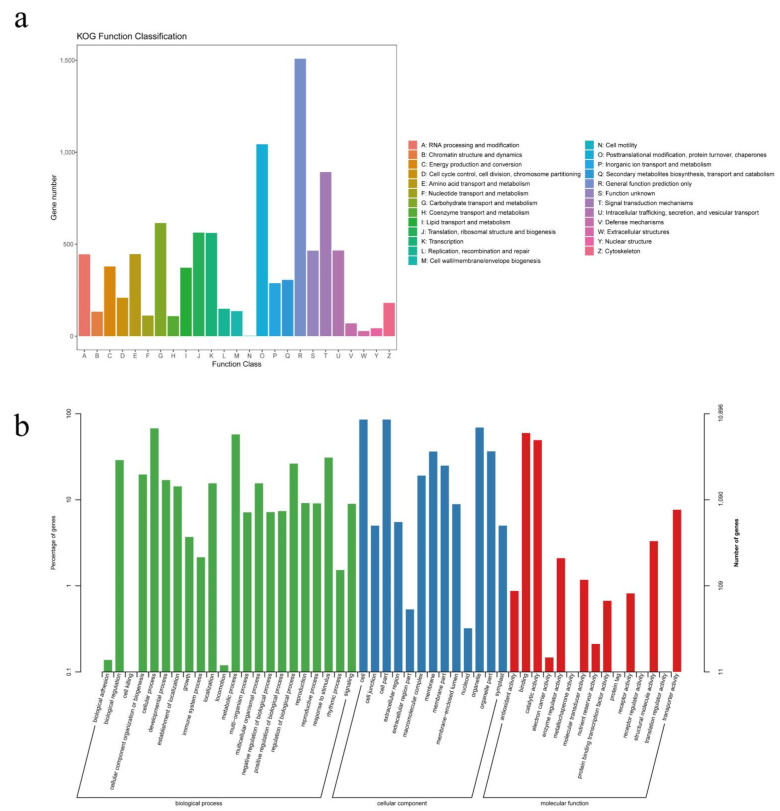
KOG (**a**) and GO (**b**) database function annotation classification statistics.

**Figure 5 plants-11-03559-f005:**
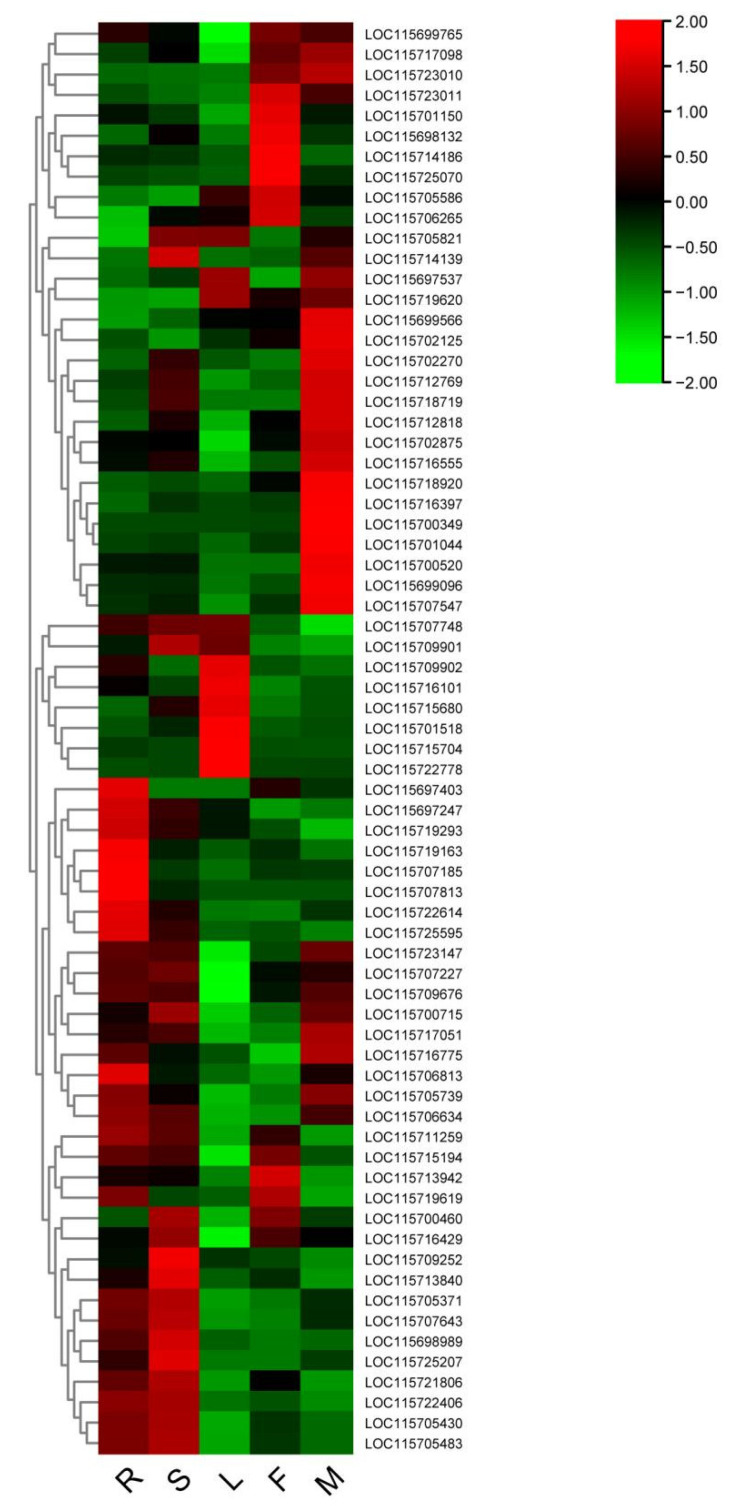
Heatmap of gene expression of genes associated with starch and sucrose metabolism in different tissues of cannabis (R: root; S: stem; L: leaves; F: female; M: male).

**Figure 6 plants-11-03559-f006:**
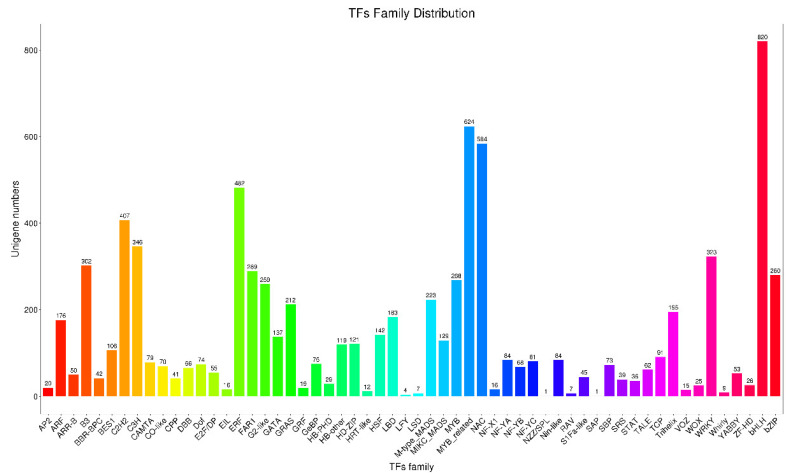
The distribution of TFs family.

**Figure 7 plants-11-03559-f007:**
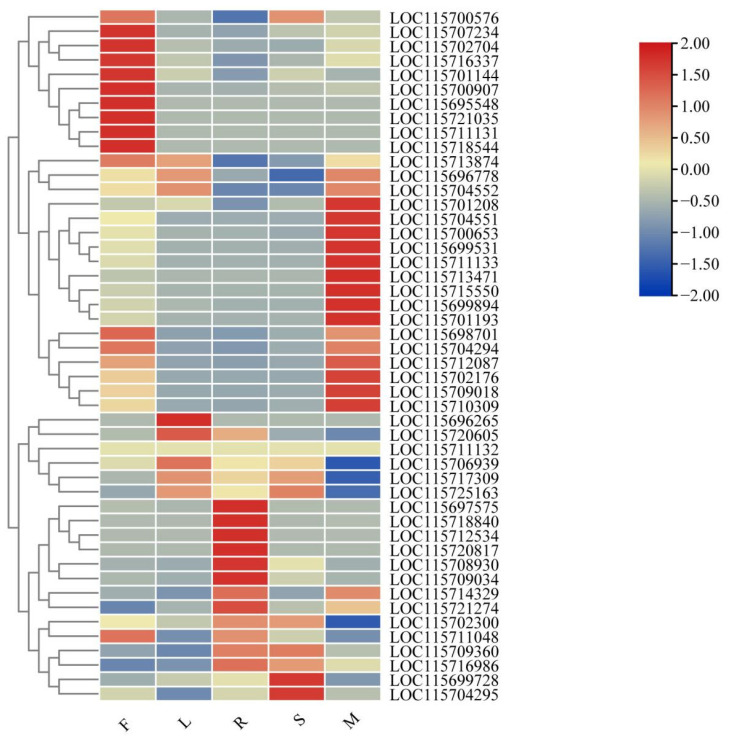
Heatmap of gene expression of MADS transcription factors in different tissues of cannabis (R: root; S: stem; L: leaves; F: female; M: male).

**Figure 8 plants-11-03559-f008:**
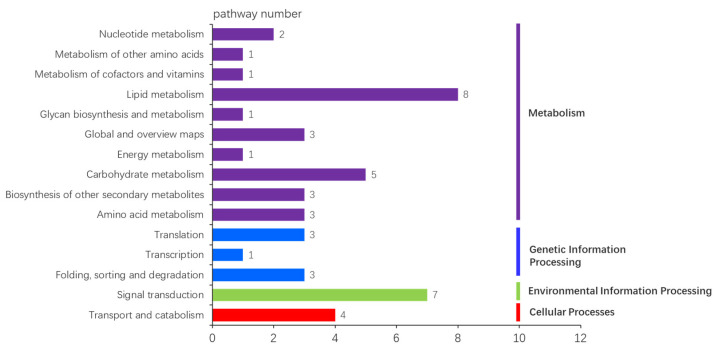
The unique pathway number of novel genes in KEGG database.

**Figure 9 plants-11-03559-f009:**
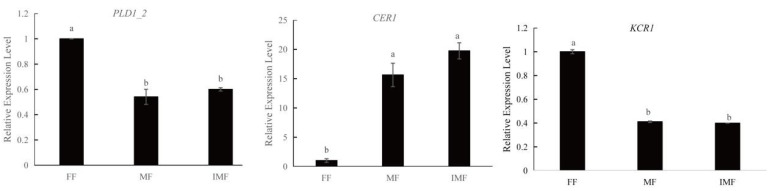
The gene expression level of three genes (*PLD1_2*, *CER1*, *KCR1*) of lipid metabolism pathway by qRT-PCR. Different letters indicate significant differences among three tissues according to one-way ANOVA test (*p* < 0.05). FF: female flower; MF: male flower; IMF: induced-male flower.

**Figure 10 plants-11-03559-f010:**
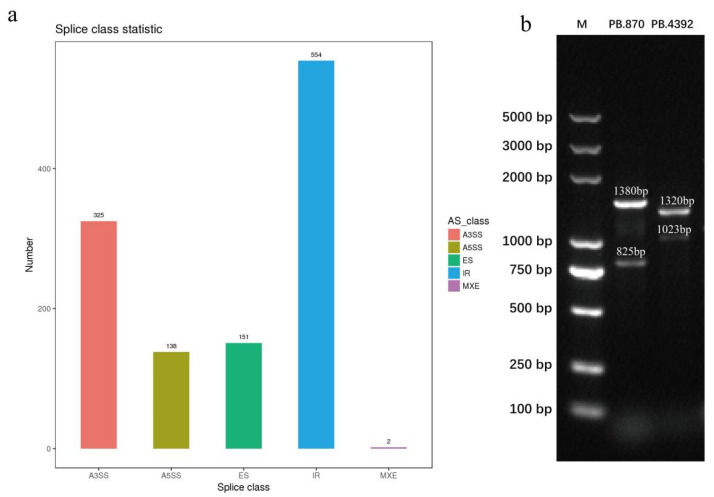
The distribution and validation of AS events. (**a**) The distribution of AS events number; and (**b**) gel electrophoresis validation of AS events PB.870 and PB.4392.

**Table 1 plants-11-03559-t001:** Statistic of CCS reads results.

Sample	CCS Reads	Full Length	FLNC Number	Percentage (%)
GYDM	347,387	301,937	290,886	83.74

**Table 2 plants-11-03559-t002:** Summary of function annotation of cannabis full-length transcriptome.

Annotated Database	Annotated Number	300 ≤ Length < 1000	Length ≥ 1000
NR	13,538 (99.13%)	1246 (9.12%)	12,288 (89.98%)
Swissprot	11,587 (84.84%)	965 (7.07%)	10,620 (77.76%)
KEGG	6025 (44.12%)	575 (4.21%)	5449 (39.90%)
KOG	8552 (62.62%)	664 (4.86%)	7887 (57.75%)
eggNOG	13,379 (97.96%)	1190 (8.71%)	12,185 (89.22%)
GO	10,896 (79.78%)	942 (6.90%)	9952 (72.87%)
Pfam	12,844 (94.05%)	1038 (7.60%)	11,806 (86.45%)

**Table 3 plants-11-03559-t003:** The top 11 pathways of function annotation in KEGG database.

Classification	Pathway Definition	Pathway	Gene Number
Metabolism–Carbohydrate metabolism	Glycolysis/Gluconeogenesis	ko00010	138
	Starch and sucrose metabolism	ko00500	173
	Amino sugar and nucleotide sugar metabolism	ko00520	141
Metabolism–Global and overview maps	Carbon metabolism	ko01200	321
	Biosynthesis of amino acids	ko01230	253
Genetic Information Processing–Translation	Ribosome	ko03010	253
	RNA transport	ko03013	148
	mRNA surveillance pathway	ko03015	125
Genetic Information Processing–Transcription	Spliceosome	ko03040	209
Environmental Information Processing–Signal transduction	Plant hormone signal transduction	ko04075	174
Genetic Information Processing–Folding, sorting and degradation	Protein processing in endoplasmic reticulum	ko04141	180

**Table 4 plants-11-03559-t004:** Summary of novel genes analysis.

Sample	Total Reads	Total Base (bp)	Min Length	Max Length	Mean Length
Novel genes	232	417,131	346	4706	1797.98

## Data Availability

Not applicable.

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
