# Peer review of "Single-Molecule Real-Time Sequencing of Full-Length Transcriptome and Identification of Genes Related to Male Development in Cannabis sativa"

_plants, 2022, doi:10.3390/plants11243559_

Round 1

Reviewer 1 Report

Developing hermaphrodites for producing feminized seeds (through chemical induction) has important bearing on the cannabis industry. In this manuscript authors attempted to genetically characterize the male development in cannabis, a commercially and medicinally important plant. To address this authors generated a single library of full-length transcriptome (Iso-seq) from pooled tissues. By annotating those transcripts using homology search against public databases, authors tried to decipher the reproductive biology. Using unpublished RNA-seq data from monoecious cultivar, authors attributed a set of MADS transcription factors to male flower development. Manuscript is well written however more details and literature review is needed about the importance of developing homozygous females. Measurement of abundance of the entire set of transcripts is missing and a time course experiment leading to the development of male flowers would be a good way to support the statements made in the manuscript. Genes selected in this study appears to be cherry-picking and not a rational attempt to understand the biology.

Minor comments:

Methods

Lines 138-140: Transition to identification of non-coding RNAs is unclear.

Results

Lines 258-264: Why that transcriptome data is not included as part of this study? To support the statements made in the narrative, RNA-seq of samples used in the Iso-seq should be done to interrogate quantitatively otherwise it is all speculative.

Lines 299-230: Where is the size of those genes from Iso-seq provided?

Author Response

Dear editor:

Thank you very much for handling our manuscript entitled “Single-Molecule Real-Time Sequencing of Full-Length Transcriptome and Identification of Genes Related to Male Development in Cannabis sativa (plants-1967740). We are also grateful to the reviewers who gave invaluable comments to our manuscript. Our study focused on SMRT sequencing of mixed samples of female and induced male flowers of cannabis, and analysis of annotated information, novel genes, transcription factors, and AS events to provide information for understanding male development of cannabis. According to the reviewer's comments, we have revised the manuscript for each problem, especially improved the manuscript results and the content of the discussion part, so as to have a more comprehensive understanding and support for the experimental results. Now we submit the revised manuscript and hope that the revised manuscript can be published in plants. 

The responses to reviewers’ comments are as following. 

Your kind re-consideration is highly appreciated.

The best regards

Sincerely yours

Gen Pan, Ph.D.

Institute of Bast Fiber Crops, Chinese Academy of Agricultural Science

China

Responses to Reviewer’s Comments

Reviewer #1 (Remarks to the Author):

Comments and Suggestions for Authors: Developing hermaphrodites for producing feminized seeds (through chemical induction) has important bearing on the cannabis industry. In this manuscript authors attempted to genetically characterize the male development in cannabis, a commercially and medicinally important plant. To address this authors generated a single library of full-length transcriptome (Iso-seq) from pooled tissues. By annotating those transcripts using homology search against public databases, authors tried to decipher the reproductive biology. Using unpublished RNA-seq data from monoecious cultivar, authors attributed a set of MADS transcription factors to male flower development. Manuscript is well written however more details and literature review is needed about the importance of developing homozygous females. Measurement of abundance of the entire set of transcripts is missing and a time course experiment leading to the development of male flowers would be a good way to support the statements made in the manuscript. Genes selected in this study appears to be cherry-picking and not a rational attempt to understand the biology.

Reply: Thanks for the comments and suggestions. We added to the literature review about the importance and practical significance of the development of homozygous female plants in production to better support the study. We have added sequences as “For example, if the purpose is to produce CBD and harvest seeds, the proportion of female plants in the population should be increased to gain high CBD and seed yield, whereas if the main purpose is to use fiber, the proportion of male plants should be increased to get more fiber with good quality.” in introduction section. In addition, we agree that the measurement of abundance of the entire set of transcripts and expression analysis of genes at the different development stages of male flowers would be a good way to support the statements made in the manuscript. However, we only focused on genes specifically expressed in male flowers and which can laid a foundation for the further study of mechanism of male flower development. Moreover, most of the genes mentioned in the manuscript were verified by transcriptome data of the cannabis monoecious strain USO14, different tissues (female flower, male flower, induced-male flower) and SMRT analysis results to explain the functions of these genes in the development of male cannabis flower.       

Additional, the genes studied for their biology were selected randomly in this study, and their expression levels were analyzed using 3 biological replicates and 3 technical replicates.

Comment#1 Lines 138-140: Transition to identification of non-coding RNAs is unclear.

Reply: Thanks for the comment. We have revised the text as “To identify long non-coding RNAs (lncRNA), transcripts with lengths greater than 200 bp and open reading frames (ORFs) greater than 300 bp were filtered out. The sequences annotated in the coding library were also screened out. LncRNAs were further screened to filter out transcripts with coding potential using four analysis tools, including the coding potential calculator (CPC), coding-non-coding index (CNCI), Pfam, and predictor of long non-coding RNAs and messenger RNAs based on an improved k-mer scheme (PLEK).” in Materials and Methods.

Comment#2 Lines 258-264: Why that transcriptome data is not included as part of this study? To support the statements made in the narrative, RNA-seq of samples used in the Iso-seq should be done to interrogate quantitatively otherwise it is all speculative.

Reply: Thanks for the suggestions. We agree that RNA-seq of samples used in the Iso-seq should be done, but the main purpose of our study was to provide full-length transcriptome data for further study of male development genes in cannabis. In addition, transcriptome data of the cannabis monoecious strain USO14 was only used to analyze the transcript levels of genes tested in this study. However, the transcriptome data of monoecious cannabis USO14 was a part of another study in our lab, and these data will be released with the publication of relevant papers, so we cannot provide it in this study.  

Comment#3 Lines 299-230: Where is the size of those genes from Iso-seq provided?

Reply: Thanks for the comments. The size of those genes were predicated by Astalavista software, and the results were presented in a GTF file.

Reviewer 2 Report

Single-molecule real-time Sequencing of Full-Length Transcriptome (SMRT) is one of powerful third-generation sequencing technologies, and widely used for accurately reconstruct expressed full-length transcripts, predict splice isoforms, and analyze transcriptome diversity. Here the manuscript plants-1967740 used SMRT to sequence the transcriptome of the female ZMZY1 variety cannabis plant bearing a chemical-induced male flower, and obtain high-quality long-read sequences, alternative splicing, and isoforms. The obtained results will enhance understanding about characteristics of the cannabis transcriptome and the molecular profile of male development. I recommend the acceptance for publication in Plants when the authors make the minor revision.

(1) The writing quality of manuscript should be improved.  

(2) In my opinion, the contents about the characteristics of transcriptome in the manuscript is descriptive, it should be further improved.

(3) Please check grammars of the sentences. For example, Line 176-178, “To obtain the full-length cannabis transcriptome of qualified RNAs from different tissues of the ZMZY1 variety were equally mixed for SMRT sequencing using the PacBio Iso-Seq platform.” Feel difficult to understand?

(4)  Several figures (Figures 4a, 6 and 10) are not clear.

Author Response

Dear editor:

Thank you very much for handling our manuscript entitled “Single-Molecule Real-Time Sequencing of Full-Length Transcriptome and Identification of Genes Related to Male Development in Cannabis sativa (plants-1967740). We are also grateful to the reviewers who gave invaluable comments to our manuscript. Our study focused on SMRT sequencing of mixed samples of female and induced male flowers of cannabis, and analysis of annotated information, novel genes, transcription factors, and AS events to provide information for understanding male development of cannabis. According to the reviewer's comments, we have revised the manuscript for each problem, especially improved the manuscript results and the content of the discussion part, so as to have a more comprehensive understanding and support for the experimental results. Now we submit the revised manuscript and hope that the revised manuscript can be published in plants. 

The responses to reviewers’ comments are as following. 

Your kind re-consideration is highly appreciated.

The best regards

Sincerely yours

Gen Pan, Ph.D.

Institute of Bast Fiber Crops, Chinese Academy of Agricultural Science

China

Responses to Reviewer’s Comments

Reviewer #2 (Remarks to the Author):

Comments and Suggestions for Authors: Single-molecule real-time Sequencing of Full-Length Transcriptome (SMRT) is one of powerful third-generation sequencing technologies, and widely used for accurately reconstruct expressed full-length transcripts, predict splice isoforms, and analyze transcriptome diversity. Here the manuscript plants-1967740 used SMRT to sequence the transcriptome of the female ZMZY1 variety cannabis plant bearing a chemical-induced male flower, and obtain high-quality long-read sequences, alternative splicing, and isoforms. The obtained results will enhance understanding about characteristics of the cannabis transcriptome and the molecular profile of male development. I recommend the acceptance for publication in Plants when the authors make the minor revision.

Reply: Thanks for the suggestion. We have made a comprehensive revision in revised manuscript.

Comment#1 The writing quality of manuscript should be improved.

Reply: Thanks for pointing this out. We made a complete revision of the writing of the manuscript and had it revised by English professionals.

Comment#2 In my opinion, the contents about the characteristics of transcriptome in the manuscript is descriptive, it should be further improved.

Reply: Thanks for pointing out this. We have refined the description of transcriptome characteristics in the transcripts to further validate our experimental results.

Comment#3 Please check grammars of the sentences. For example, Line 176-178, “To obtain the full-length cannabis transcriptome of qualified RNAs from different tissues of the ZMZY1 variety were equally mixed for SMRT sequencing using the PacBio Iso-Seq platform.” Feel difficult to understand?

Reply: Thanks for the suggestion. We have revised the text as “The qualified RNAs from different tissues of the ZMZY1 variety were equally mixed, and the SMRT sequencing was performed using the PacBio Iso-Seq platform to obtain the full-length transcriptome of cannabis.”. In addition, we checked the grammar in the throughtout the manuscript and made corrections.

Comment#4 Several figures (Figures 4a, 6 and 10) are not clear.

Reply: Thanks for the suggestion. We have replaced the figures in the revised manuscript.

Reviewer 3 Report

The manuscript by Jiang et al. aimed at the identification of genes responsible for male development in Cannabis sativa by single-molecule real-time sequencing of full-length transcriptome. I think that although the topic could be considered as innovative, the manuscript seems to me to be too descriptive and I somehow lack a more sound biological relevance behind. I also did not like the discussion that represented for me rather a repetition of results section with some references to previous studies. The discussion should be greatly improved.

I would also re-consider the number of tables and figures in the manuscript. I think that several of them could be merged together to improve the readability of the manuscript. 

Then, I have these two minor comments:

lines 58-61 - It should be stated there, that the gene names are addressing Arabidopsis thaliana, in other species, there are sometimes different names used (for instance in snapdragon). Mentioning sepalata only is not exact, I would mention all its 3 homologues, i.e. sepalata 1/2/3.

line 401-408 - The authors did not provide with the specification of their contributions.

Author Response

Dear editor:

Thank you very much for handling our manuscript entitled “Single-Molecule Real-Time Sequencing of Full-Length Transcriptome and Identification of Genes Related to Male Development in Cannabis sativa (plants-1967740). We are also grateful to the reviewers who gave invaluable comments to our manuscript. Our study focused on SMRT sequencing of mixed samples of female and induced male flowers of cannabis, and analysis of annotated information, novel genes, transcription factors, and AS events to provide information for understanding male development of cannabis. According to the reviewer's comments, we have revised the manuscript for each problem, especially improved the manuscript results and the content of the discussion part, so as to have a more comprehensive understanding and support for the experimental results. Now we submit the revised manuscript and hope that the revised manuscript can be published in plants. 

The responses to reviewers’ comments are as following. 

Your kind re-consideration is highly appreciated.

The best regards

Sincerely yours

Gen Pan, Ph.D.

Institute of Bast Fiber Crops, Chinese Academy of Agricultural Science

China

Responses to Reviewer’s Comments

Reviewer #3 (Remarks to the Author):

Comments and Suggestions for Authors: The manuscript by Jiang et al. aimed at the identification of genes responsible for male development in Cannabis sativa by single-molecule real-time sequencing of full-length transcriptome. I think that although the topic could be considered as innovative, the manuscript seems to me to be too descriptive and I somehow lack a more sound biological relevance behind. I also did not like the discussion that represented for me rather a repetition of results section with some references to previous studies. The discussion should be greatly improved. I would also re-consider the number of tables and figures in the manuscript. I think that several of them could be merged together to improve the readability of the manuscript.

Reply: Thanks for the suggestions. Our study mainly describes the obtained full-length transcriptome data of cannabis to provide data support for the study of the male development mechanism of cannabis via male-special expression genes. Therefore, the content of discussion is mainly to prove that these genes may play a role in male development of cannabis based on previous studies, so as to provide more comprehensive genetic information for future studies. According to the suggestions, we revised the discussion section partly. In addition, we had merged some figures together to improve the readability of the manuscript. For example, figure 10 and figure 11 were merged together in the revised manuscript.

Comment#1 lines 58-61 - It should be stated there, that the gene names are addressing Arabidopsis thaliana, in other species, there are sometimes different names used (for instance in snapdragon). Mentioning sepalata only is not exact, I would mention all its 3 homologues, i.e. sepalata 1/2/3.

Reply: Thanks for pointing this out. To aviod wrong statement of gene names controlling the development of flower, we revised the lines 58-61 as “The key responsive genes reported previously in Arabidopsis thaliana include the A-class genes AP1 (APETALA1) and AP2 (APETALA1), the B-class gene PISTILLATA (PI), the C-class gene AGAMOUS (AG), the D-class genes FBP7 (Floral Bind 7) and FBP1 (Floral Bind 1), and the E-class gene SEPALLATA (SEP) .

Comment#2 line 401-408 - The authors did not provide with the specification of their contributions.

Reply: Thanks for the comments. We have added the specification of the author's contribution to the manuscript as follows: “Conceptualization, Siqi Huang and Guang Yang; Methodology, Mingbao Luan and Guang Yang; Validation, Hui Jiang and Ying Li; Investigation, Ying Li; Resources, Mingbao Luan, Siqi Huang, Lining Zhao and Gen Pan; Data curation, Hui Jiang and Ying Li; Writing–original draft, Hui Jiang; Writing–review & editing, Gen Pan; Project administration, Lining Zhao, Guang Yang and Gen Pan; Funding acquisition, Gen Pan. All authors have read and agreed to the published version of the manuscript.

Round 2

Reviewer 3 Report

Thank you for improving the manuscript. I am happy with it now.